# Tracheostomy care quality improvement in low- and middle-income countries: A scoping review

**Msiba Selekwa**[1‡], **Ivy Maina**[2‡], **Tiffany Yeh**[3], **Aslam Nkya**[4], **Isaie Ncogoza**[5], **Roger C. Nuss**[6], **Beatrice P. Mushi**[1], **Sumaiya Haddadi**[1], **Katherine Van Loon**[7], **Elia Mbaga**[1], **Willybroad Massawe**[4,8], **David W. Roberson**[9,10], **Nazima Dharsee**[1,11], **Baraka Musimu**[8], **Mary Jue Xu**[12,13]*

1 Muhimbili University of Health and Allied Sciences, Dar es Salaam, Tanzania, 2 Department of Otolaryngology-Head and Neck Surgery, Perelman School of Medicine, University of Pennsylvania, Philadelphia, PA, United States of America, 3 Perelman School of Medicine, University of Pennsylvania, Philadelphia, PA, United States of America, 4 Department of Otolaryngology-Head & Neck Surgery, Muhimbili National Hospital, Dar es Salaam, Tanzania, 5 Department of Surgery, College of Medicine & Health Sciences, University of Rwanda, Kigali, Rwanda, 6 Department of Otolaryngology and Communication Enhancement, Boston Children's Hospital, Boston, Massachusetts, United States of America, 7 Department of Medicine, Division of Hematology/Oncology, University of California San Francisco, San Francisco, California, United States of America, 8 Department of Otolaryngology-Head and Neck Surgery, Muhimbili University of Health and Allied Science, Dar es Salaam, Tanzania, 9 Bayhealth Medical Group, Dover, Delaware, United States of America, 10 Global Tracheostomy Collaborative, Raleigh, North Carolina, United States of America, 11 Ocean Road Cancer Institute, Dar es Salaam, Tanzania, 12 Department of Otolaryngology-Head and Neck Surgery, University of California San Francisco, San Francisco, California, United States of America, 13 National Clinician Scholars Program, University of California San Francisco, San Francisco, California, United States of America

‡ MS and IM are co-first authors on this work.
* maryjue.xu@ucsf.edu

**Data Availability Statement:** All data collected available in the manuscript.

**Funding:** MJX is funded by the National Clinician Scholars Program, but the program did not fund

## Abstract

Tracheostomy is a lifesaving, essential procedure performed for airway obstruction in the case of head and neck cancers, prolonged ventilator use, and for long-term pulmonary care. While successful quality improvement interventions in high-income countries such as through the Global Tracheostomy Collaborative significantly reduced length of hospital stay and decreased levels of anxiety among patients, limited literature exists regarding tracheostomy care and practices in low and middle-income countries (LMIC), where most of the world resides. Given limited literature, this scoping review aims to summarize published tracheostomy studies in LMICs and highlight areas in need of quality improvement and clinical research efforts. Based on the PRISMA guidelines, a scoping review of the literature was performed through MEDLINE/PubMed and Embase using terms related to tracheostomy, educational and quality improvement interventions, and LMICs. Publications from 2000–2022 in English were included. Eighteen publications representing 10 countries were included in the final analysis. Seven studies described baseline needs assessments, 3 development of training programs for caregivers, 6 trialed home-based or hospital-based interventions, and finally 2 articles discussed development of standardized protocols. Overall, studies highlighted the unique challenges to tracheostomy care in LMICs including language, literacy barriers, resource availability (running water and electricity in patient

this study. The authors received no specific funding for this work. The funders had no role in study design, data collection and analysis, decision to publish, or preparation of the manuscript.

**Competing interests:** The authors have declared that no competing interests exist.

homes), and health system access (financial costs of travel and follow-up). There is currently limited published literature on tracheostomy quality improvement and care in LMICs. Opportunities to improve quality of care include increased efforts to measure complications and outcomes, implementing evidence-based interventions tailored to LMIC settings, and using an implementation science framework to study tracheostomy care in LMICs.

## Background

Tracheotomy is an essential, life-saving procedure worldwide associated with potentially serious complications [1]. An estimated 250,000 procedures are performed annually in high income countries (HICs) [2–5]. Although a relatively simple procedure, tracheostomy involves risks and complications such as oxygen desaturation during the procedure, peristomal and tracheal bleeding, and accidental decannulation which all carry risk of mortality. Published literature from both HIC and low and middle-income country (LMIC) settings reports complication rates ranging from 4% to 66% and all-cause mortality rates up to 22% [3, 4, 6]. The observed wide variation in complication rates are attributed to differences in practices, resources, and regulatory policies. Particularly in LMICs where health systems are overburdened and fragile, complications following tracheotomy may lead to increased readmissions, long hospital stays, and unnecessary costs to both patients and healthcare facilities.

While quality improvement (QI) interventions for tracheostomy care in HICs have led to significant improvements, less has been studied in LMIC health settings, where the majority of the global population resides. The Global Tracheostomy Collaborative (GTC) was established in 2012 to champion successful interventions in improving safety and care among tracheostomized patients [7]. The GTC published five key drivers of QI which include: i) joint decision-making through multidisciplinary care, ii) establishing standardized care protocols, iii) staff education, iv) patient and family involvement in improvement efforts and v) establishing and utilizing an outcome-based metrics database. GTC member hospitals have reported significant reductions in length of hospital stay, time on the ventilator, and levels of anxiety among patients after implementing these key drivers [8]. To date, however, few comprehensive interventions as that proposed by the GTC exist in LMICs where issues such as resource-constraints, language barriers, and literacy are pressing, unique challenges.

This study reviews the literature on tracheostomy care in LMICs. Given the limited number of publications and heterogeneous nature of the studies on tracheostomy care in LMICs, a scoping review was chosen. Herein, we aim to summarize the existing data and highlight gaps in knowledge for future research.

## Methods

### Study design

This study followed the PRISMA Guidelines for Scoping Reviews (S1 Checklist) [9].

### Search strategy

The literature search was implemented on December 20, 2022 in two electronic databases: MEDLINE/PubMed and Embase. The query combined search strings on tracheostomy, key terms related to developing countries, and individual LMICs as defined by the 2022 World Bank income classification (S1 Appendix). The search was narrowed to publication years

2000–2022. A filter for full text was applied in PubMed, and a filter for articles and reviews was applied in Embase. Abstracts were excluded in PubMed, and conference abstracts, letters, conference reviews, articles in press, conference papers, editorials, notes, short surveys, errata, and tombstones were excluded in Embase. Non-English language studies were excluded.

### Article selection

Citations were imported into the reference manager EndNote 20 and duplicate citations were removed. Publications underwent title and abstract relevance screening by one author (IM). Final papers for relevance were reviewed and confirmed by two authors (IM, MJX). The criteria for exclusion included articles which did not feature patients with tracheostomies or clinicians/caregivers caring for patients with tracheostomies, articles not published in English, and/or articles that did not have full text available.

### Data characterization

Variables for data abstraction were composed and reviewed by the research team. The variables spanned the categories of paper variables (first author and year, country); intervention (intervention objective, patient population, number of participants, language, training objectives, training duration, background of learners and educators, educational materials, equipment provided, post-discharge support, patient socioeconomic demographics, relationship of primary caregiver to patient); outcomes (patient outcomes, intervention evaluation, cost); and scale (sustainability and dissemination). Data were abstracted by one author (IM) and confirmed by a second author (MJX).

## Results

### Article selection

A total of 1624 search results were identified (MEDLINE/PubMed 573 and Embase 1051, Fig 1). After duplicate removal, 1493 publications underwent subsequent title and abstract relevance screening with 1437 excluded based on non-relevance or non-applicability, 5 publications were excluded due to their formats as abstracts or editorial commentaries. Based on relevance to training and services for tracheostomy tube care in LMICs, 18 publications were selected for final analysis.

### Study characteristics

Of the eighteen articles, six (33%) described programs in Africa, five (28%) in Asia, four (22%) in South America, and three (17%) in transcontinental Turkey. Eight articles (44%) studied pediatric patients exclusively, nine (50%) adults, and one (6%) examiend both populations. The studies were divided into the following thematic categories: (1) needs assessments for tracheostomy care programs; (2) program descriptions; (3) tracheostomy care interventions; and (4) formulating tracheostomy care protocols.

### Needs assessment for tracheostomy care programs

Seven studies on needs assessments for tracheostomy care programs highlighted evaluation of tracheostomy care knowledge among adult intensive care unit (ICU) nurses, experience of caregivers following discharge from the hospital, and exploration of barriers related to social integration for patients. (Table 1) [10–16].

Focusing on knowledge of ICU nursing staff, Mungan et al. (2019) surveyed 138 ICU nurses at three tertiary hospitals in Turkey and noted a lack of knowledge for emergency

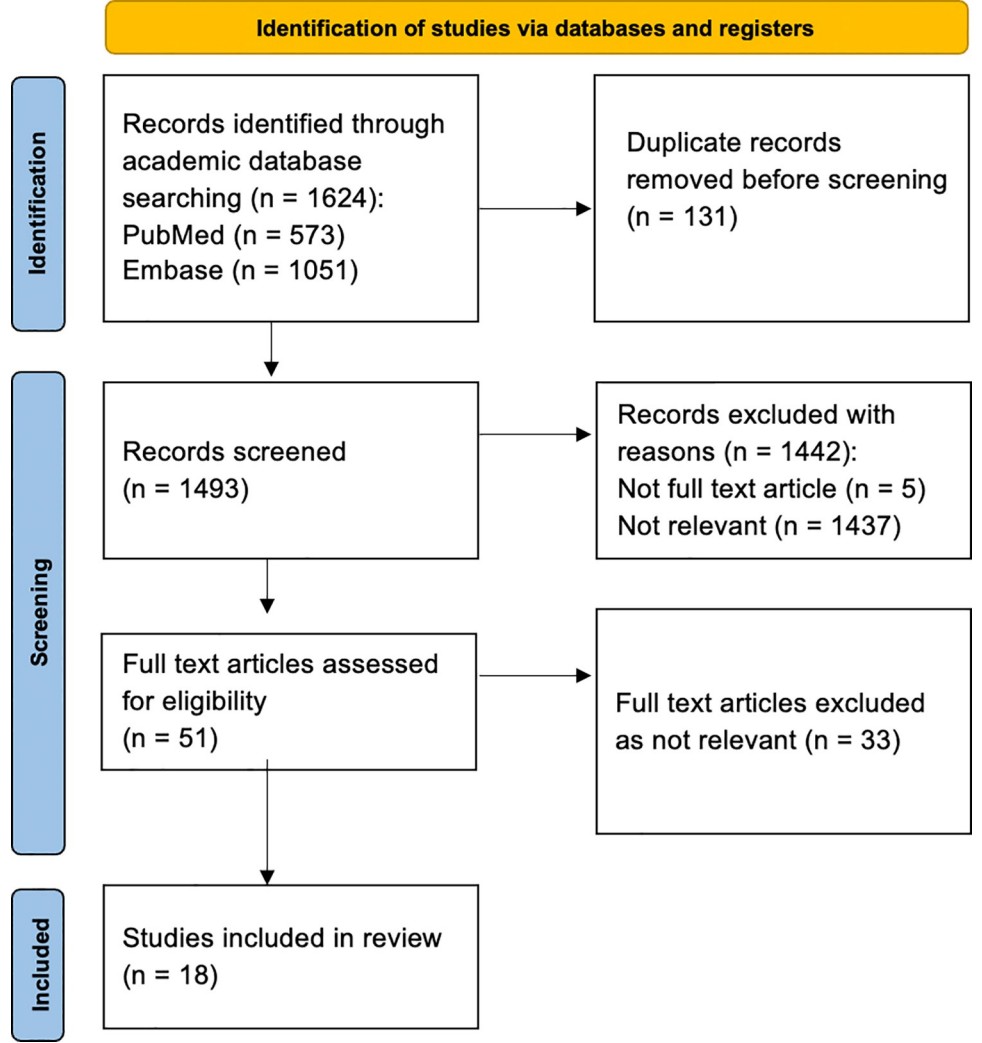

**Fig 1. PRISMA diagram for study inclusion.**

tracheostomy care management with questions answered correctly by 30% of respondents [16]. Lack of knowledge was not associated with the nursing qualifications nor time in the ICU. The authors concluded that knowledge-specific assessments may be more revealing than self-assessments of confidence level and that standardized care education was a critical gap in their system.

The next set of needs assessment studies focused on caregiver experiences during and after hospitalization for adult and pediatric patients [10–13]. Common needs that emerged across the studies despite patient age were 1) inadequate pre-procedural counseling, 2) need for more training prior to discharge home, 3) lack of support at home following hospitalization, 4) financial burden of caregivers, and 5) the psychological burden of caregivers. Proposed recommendations included structured training and protocols for caregivers, home-based post-discharge support programs, expanded resources such as online videos, and finally caregivers connecting to broader communities through online forums for support and to reduce social isolation. None of the studies specifically made recommendations for mitigating financial burden.

**Table 1. Tracheostomy care needs assessment studies.**

| General Theme | Hospital Care Provider Knowledge | Caregiver Knowledge and Support | | | | Patient Social Integration Following Hospitalization | |
|---|---|---|---|---|---|---|---|
| First Author, Year | Mungan 2019 | Daraie 2021 | de Lemos 2019 | Karaca 2019 | Gong 2018 | Mahomva 2016 | Akenroye 2013 |
| Country | Turkey | Iran | Brazil | Turkish | China | South Africa | Nigeria |
| Population | Adult | Adult | Pediatric | Adult | Pediatric | Pediatric | Adult |
| Study Design | Cross sectional | Qualitative | Qualitative | Cross sectional | Qualitative | Qualitative | Case series |
| Language | Not specified | Not specified | Not specified | Not specified | Not specified | Zulu | Not specified |
| Study Objective | Evaluate tracheostomy care knowledge among intensive care unit (ICU) nursing staff | Assess caregiver experiences and challenges | Describe social support for families of tracheostomized children | Describe care practices and burden on family caregivers of adults patients with tracheostomies | To describe the experiences of caregivers of children with tracheostomies during the first month after transition from hospital to home | Describe barriers of children with tracheostomies attending school. Devise tools for teachers and administrators to facilitate acceptance and safety in classrooms. | Assess the impact of permanent tracheostomy on surgical/medical complications and social integration |
| Study Participants | 138 ICU nurses from three tertiary hospitals | 9 adult family caregivers and 1 professional caregiver | 9 adult family caregivers | 50 adult family caregivers | 13 adult family caregivers | 4 pediatric patients 4 primary caregivers 4 primary teachers | 4 patients with bilateral laryngeal nerve paralysis following thyroidectomy |
| Format of Needs Assessment | Survey of demographics and knowledge | In-depth unstructured qualitative interview | Semi-structured qualitative interview and closed item questionnaire | Survey of patient care practices and caregiver burden | Semi-structured qualitative interview | Semi-structured, qualitative interview and closed item questionnaire | Informal; based on observation of complications and perceived social setbacks |
| Identified Needs/ Barriers | • Majority of nurses expressed confidence in tracheostomy care <br>• Questions on routine tracheostomy care answered with greater frequency than questions on emergency tracheostomy care | • Inadequate training prior to discharge <br>• Inadequate support for care of patients at home <br>• High psychological and financial burden | • Inadequate pre-procedure counseling resulted in caregivers feeling unprepared <br>• Caregiver training was overwhelming and rushed <br>• Anxiety about tracheostomy care and social isolation after discharge <br>• High financial burden | • Caregiver burden higher for caregivers who were female, with chronic illnesses, without social support, caring for patients without health insurance <br>• Burden increased with increased daily care time and for caregivers required to perform daily nebulization, deliver oxygen therapy or clean external cannulas | • Lack of professional support after discharge <br>• High psychological burden due to anxiety, depression and social isolation | • Teacher barriers: lack of knowledge about tracheostomies, unclear liability for safety issues <br>• Safety barriers: lack of running water, distant pit latrines, environmental contaminants (i.e. sand) <br>• Needs: hand hygiene, materials, equipment availability (extra trach, suction), teacher training | • Stigmatization due to visibility of tracheostomy tube <br>• Scarcity of speaking valves <br>• Familial acceptance and willingness to receive training for tracheostomy care |

*(Continued)*

**Table 1.** (Continued)

| General Theme | Hospital Care Provider Knowledge | Caregiver Knowledge and Support | | | | Patient Social Integration Following Hospitalization | |
|---|---|---|---|---|---|---|---|
| **Proposed/ Enacted Measures to Address Needs/ Barriers** | • Assessment of nursing education and knowledge<br>• Development of standardized education program for all clinicians/ caregivers | • Proposed establishment or contribution of non-governmental organizations dedicated to supporting the home care of patients with tracheostomies | • Structured and graduated caregiver training protocols<br>• Simulations for tracheostomy emergencies<br>• Online information and forums (e.g. blogs, social media networks) to increase confidence and reduce social isolation<br>• Videos and care protocols | • Development and expansion of home care services to provide support to caregivers | • Training and education programs should be developed for caregivers<br>• Professional support in the form of instructional written protocols and videos | • Information booklet for teachers<br>• Recommendations for inclusion of students in the school system developed with and delivered to key stakeholders<br>• Training program and collaboration between hospital staff, Department of Education and families was proposed | • Conferences between care team and families during hospitalization to educate families<br>• Improvisation of neck covers hiding tracheostomy tubes<br>• Improvised fenestrations drilled into available tracheostomy tubes to allow for voice use |

The final set of needs assessments focused on social integration of patients with tracheostomies. A qualitative study by Mahomva et al. (2016) on barriers to school attendance for children in South Africa led to the development of an introductory booklet for teachers, development of recommendations for integration of children with tracheostomies into school, and a recommendation for a more formalized collaboration between families, the department of education, and hospital staff [15]. In Nigeria through observational data on adults following iatrogenic vocal fold paralysis, Akenroye et al. (2013) noted challenges with social integration and stigmatization due to tracheostomy visibility and communication barriers from lack of speaking valves [14]. Improvisations such as neck covers to reduce tracheostomy visibility and placing fenestrations into non-fenestrated tracheostomy tubes were innovative, resource-conscious solutions. These two studies overall emphasized the importance of supporting patient re-integration into society in LMICs.

## Program descriptions

Three studies described training programs for caregivers of pediatric patients with tracheostomies and reported retrospectively collected complication rates (Table 2) [17–19]. All studies reported challenges obtaining home-based tracheostomy care equipment, including spare tracheostomy tubes, suction catheters, and suction pumps. Manually operated suction pumps were mentioned by all articles and favored over electronic pumps, considering resource-constraints of families. Across studies, all-cause mortality ranged from 10–21% and tracheostomy-related mortality ranged from 1.2–2.4% following discharge.

Studies reported notable findings related to readmission rates and the social needs of patients. Vanker et al. (2010) and Groenendijk et al. (2016) reported readmission rates of 95% and 50%, respectively [18, 19]. This high rate included planned readmissions for procedures. Aside from scheduled cases, infection was a common cause for readmission, leading to 34–75% of the readmissions. Finally, social factors were highlighted in both studies. In Vanker et al., 745 social work interactions were noted for 56 patients, and in Groenendijk et al., caregiver substance use was associated with a 2.6-fold increased risk for unplanned readmission.

**Table 2. Tracheostomy care program descriptive studies.**

| First Author, Year | Zia 2010 | Vanker 2010 | Groenendijk 2016 |
|---|---|---|---|
| Country | Pakistan | South Africa | South Africa |
| Population | Pediatric | Pediatric | Pediatric |
| Study Design | Retrospective review | Retrospective review | Retrospective review |
| Study Objective | Describe indications and complications of tracheostomy and the feasibility of home care for tracheostomy patients | Describe the tracheostomy home program at a tertiary children's hospital in a low resource setting | Describe the Breatheasy Tracheostomy Program (Red Cross Hospital, Cape Town) <br> Assess the impact of socio-economic circumstances on success of home care |
| Study Participants | 81 patients discharged home with tracheostomy tubes (out of 127 patients with tracheostomy performed) | 56 patients discharged with tracheostomy tubes <br> 47, 83% discharged to home <br> 9, 16% to institutional care | 157 patients discharged home with tracheostomy tubes |
| Training Initiatives | Hospital staff trained primary caregivers on tracheostomy tube changing and care including suctioning | Primary caregiver trained by specialist nursing team for a minimum of 2 weeks | Advanced pediatric clinical nurse counsels primary caregiver/family. <br> Training begins when patient medically stable; training includes tracheostomy tube changing, suctioning, humidification, basic life support skills. Manual on ventilator principles/techniques provided if needed. |
| Equipment | Spare tracheostomy tube <br> Suction catheters <br> Foot-operated suction pump | Foot-operated suction pump | Spare tracheostomy tubes <br> Manually or electrically operated suction pump |
| Post-Discharge Support | Outpatient follow-up clinic | Not specified | Follow up with advanced pediatric clinical nurse specialists |
| Outcomes for Patients Discharged with Tracheostomies | Mortality in hospital <br> • Tracheostomy-related 0.7% <br> • Unrelated to tracheostomy 18.1% <br> Mortality post-discharge <br> • Tracheostomy-related 2.4% <br> • Unrelated to tracheostomy 7.4% <br> Tracheostomy related complication rate post-discharge 22% <br> · Accidental decannulation 1.2% <br> · Cannula obstruction 5% <br> · Tracheocutaneous fistula 5% <br> Decannulation rate 49% <br> Lost to follow up 41% | Overall survival rate: <br> • 82% <br> • Discharge to home (85%) vs institution (66%), p = 0.19 <br> Readmission rate overall 95% <br> Median number of readmissions 3–4 times/child <br> Reasons for readmission <br> · Infection (34%) <br> · Tracheostomy related complications (7%) <br> · Social indications (3%) <br> · Planned surgery/endoscopy (49%) <br> Overall decannulation rate 50% | Mortality <br> • Any cause 21.1% <br> • Tracheostomy-related mortality 1.2% <br> Readmission rate <br> · 50% of participants readmitted 1–5 times <br> · Median 2 readmissions per child <br> Reasons for readmission <br> · 75% of readmissions were for infection <br> Factors associated with readmission <br> · Association with caregiver substance use, cigarette smoking <br> · No association with housing type and caregiver education level |

## Tracheostomy care interventions

Six studies assessed home and hospital-based tracheostomy care interventions (Table 3) [20–25].

Two studies assessed home-based interventions for postoperative neurosurgical patients in India and Pakistan; among those populations, a proportion (23% and 35%, respectively) were patients with tracheostomy tubes. In a randomized-control trial in India comparing standard paper-based discharge education with supplemented mobile app education and mobile messaging to the hospital team, patients and caregivers in the experimental group reported significantly higher confidence with tracheostomy tube suctioning, higher satisfaction with discharge training, and significant preference for the mobile-based discharge training [25]. In Pakistan, Khan et al. (2016) examined whether care differed between families who elected to have at-home nursing care after discharge compared to care by family caregivers alone [24]. The study found no difference in time to decannulation or 30-day mortality with the addition of a trained nurse visiting on an outpatient basis. These findings were in the context of the patient care coordinator actively evaluating the home care needs and additional family input on whether a nurse was needed.

**Table 3. Tracheostomy care intervention studies.**

| General Theme | Home Care | | Clinician and Hospital-Based Care | | | |
|---|---|---|---|---|---|---|
| First Author, Year | Metilda 2021 | Khan 2016 | Sandler 2020 | Bayram 2019 | de Carvalho 2008 | Chiaravalli 2017 |
| Country | India | Pakistan | Rwanda | Turkey | Brazil | Malawi |
| Population | Adult | Adult | Pediatric | Adult | Adult | Adult |
| Study Design | Randomized controlled trial | Retrospective, cohort | Pre- and post-training study | Randomized controlled trial | Pre- and post-training study | Simulation and evaluation |
| Training Language | English, Hindi | Not applicable | English, French | Turkish | Not specified | Not specified |
| Study Objective | Assess a supplemental mobile-app home care program (tracheostomy care was a component) | Compare outcomes of post-op neurosurgery home care by family caregivers versus professional nurse | Train clinicians on routine tracheostomy care and train patients/caregivers prior to discharge. Create locally-sourced tracheostomy home kits for patients | To determine the effect of a game-based virtual reality phone application on skills related to tracheostomy care | Determine the efficacy of nursing education program for routine tracheostomy care | Develop and assess a simulation of an intensive care unit-based tracheostomy, airway emergency |
| Study Participants | 100 post-operative neurosurgical patients 50 patients per arm 23 tracheostomy patients | Nurse group = 94 patients Family group = 102 patients 35% of all patients required trach care | Model: Train the trainer Educators: Nurse educator from Boston Children's Hospital (USA) Learners (number): Hospital-based clinicians (10) | 86 nursing students | 100 nursing personnel; 53 nursing assistants, 36 nursing technicians, 21 nurses at an oncology hospital | Model: Train the trainer Educators: Local clinicals (2 per session) Learners: Intensive care unit nurses, nursing students, clinical officers |
| Intervention | Components: Control: routine paper-based discharge education, outpatient follow-up clinic Intervention: control with additional mobile app–based discharge teaching, routine follow-up with additional remote communication through mobile app | Components: Patient care coordinator (PCC) identifies nursing needs and support system. Family decides upon trained attendant (family member) or professional nurse. Nurses and physical therapists train family member. PCC assesses care quality and approves discharges when satisfactory. | Components: 1. Low fidelity simulated trach care training course for clinicians to train patients and caregivers: 2. Tracheostomy education booklet for patients and caregivers for outpatient care: image-based, language- and literacy-independent | Components: All students: 1 hour presentation, 1 hour demonstration, 1.5 hour small group practice sessions. Experimental group: Additional tracheostomy care game-based virtual reality phone application for 7 days. | Components: Educational session conducted in small groups covering aspects of routine tracheostomy care including tracheal aspiration technique, stoma care, and usage of sterile technique Duration: 8 hours | Components: Low-fidelity simulation of a deteriorating patient in the intensive care unit Duration: 10–15 min simulated scenario, followed by 20–30 min debrief |
| Equipment | Mobile app Aimeo: • Developed by HealthCius Services (India) • Free, English and Hindi • Individualized discharge summary and educational videos, SMS reminders • Ability to contact hospital with queries and upload media • Caregivers enter vitals for remote monitoring by hospital personnel | Not specified | Locally sourced, discharge kits • 1 pamphlet (4.97 USD) • 1 manually operated DeLee suction catheter (1.80 USD) • 5 pairs of gloves (0.33 USD) • 10 saline burettes (0.90 USD) • 2 gauze rolls (0.65 USD) • 12 lubrication packets (0.72 USD) • 2 pipe cleaners (0.34 USD) • 1 humidification device (donated) • 1 trach tie (donated) • 1 trach sponge (donated) | Mobile virtual reality application featuring a simulated nurse and patient encounter requiring preparing the correct materials for tube suctioning, inner canula cleaning, and peristomal skin care. | PowerPoint presentation | Simulation course materials • Mannequin (Life/form Tracheostomy Care Simulator) • Stethoscope • Story board and summary sheet for facilitators • Oxygen, suction, and tracheostomy care equipment • PPE |

*(Continued)*

**Table 3.** (Continued)

| General Theme | Home Care | | Clinician and Hospital-Based Care | | | |
|---|---|---|---|---|---|---|
| **First Author, Year** | **Metilda 2021** | **Khan 2016** | **Sandler 2020** | **Bayram 2019** | **de Carvalho 2008** | **Chiaravalli 2017** |
| **Evaluation** | • Data collected at baseline • First follow-up within 30 days • Second follow-up within 60 days | Assessment during routine outpatient follow-up clinic | Pre- and post-training course self-efficacy questionnaire | Written knowledge test and Objective Structured Clinical Exam before and 7 days later post-intervention. | Questionnaire pre- and post-education session | Expected actions checklist during simulation. Post-scenario survey by learners on change in knowledge and confidence, simulation effectiveness/ design, feedback techniques, instructor preparedness. |
| **Outcomes** | Confidence performing suctioning: higher in experimental group (visit 1 p = 0.03, visit 2 p = 0.05) Revisits for complications: 27% overall; 32% control, 22% experimental (p = 0.26) Satisfaction with discharge teaching: no significant difference at visit 1, experimental group with higher satisfaction at visit 2 (p = 0.001) Satisfaction with discharge teaching by modality: experimental group preferred mobile app to paper-based discharge (p = 0.003) | No statistically significant difference in time to decannulation (nursing = 23.3 days, family = 19.7 days) nor 30-day mortality (nursing = 18%, family = 13%, p = 0.30) | Median confidence levels on 10-pt Likert scale: Suctioning (pre 8, post 10) Changing ties (pre 8, post 10) Tube exchange (pre 5, post 9) Guide caregivers through educational material (pre 7, post 10) Emergency situations (pre 6.5, post 10) Post-educational module survey on perceived usefulness of take-home educational resources based on 10-pt Likert scale (mean): Image-based pamphlet 9.5 Ambulatory supply kit 9.6 Manual suctioning device 9.6 Feedback on image-based trach pamphlet: 90% participants said easy to understand and no missing procedures | Significantly higher peristomal skin care performance scores for the experimental compared to control group. No significant difference in written knowledge test scores, tracheostomy suctioning skills or inner cannula cleaning skills between groups | Significant improvement in knowledge regarding use of protective equipment and suctioning technique. | Outcomes not published. |
| **Intervention Costs** | Not specified | Not specified | 16 USD per supply kit, donations for the pilot study | Not specified | Not specified | Not specified |
| **Educational Materials** | Mobile app publicly available for download | Not provided | Published on OPENPediatrics | Not provided | Not provided | Included in journal article publication |

Four studies assessed hospital-based interventions. Chiaravalli et al. (2017) described an ICU-based emergency airway simulation for ICU nurses and clinical officers in Malawi using a low-fidelity simulation model; no outcomes were reported [21]. Two studies focused on the education of nursing staff on routine tracheostomy care. In Brazil, de Carvalho et al. (2008) noted improved knowledge following PowerPoint-based didactics [20]. In Turkey, Bayram et al. (2019) randomized nursing students to either didactic and small groups interactive sessions or to an experimental group with an additional virtual reality, game-based tracheostomy care phone application for 1 week [22]. While written test scores and skills such as suctioning

**Table 4. Tracheostomy care protocol development studies.**

| First Author, Year | Soares 2018 | Avelino 2017 |
|---|---|---|
| Country | Brazil | Brazil |
| Patient Population | Adult and Pediatric | Pediatric |
| Study Design | Modified Delphi method | Consensus development panel |
| Study Objective | To create a multidisciplinary protocol for tracheostomy care in adult and pediatric patients including a basic home care manual and emergency management guidelines for caregivers and patients | To establish national guidelines and standards of care for tracheostomized children based on the opinion of a group of experts |
| Experts | 20 graduate level professionals with 5+ years of experience in their field and who work with patients with tracheostomies Included specialties: intensive care physicians and nurses, oncology emergency physicians, physiotherapists, cancer surgeons, head and neck surgeons, pediatric surgeons, thoracic surgeons, operating room nurses, nurses, endoscopy nurses | Otorhinolaryngologists with proven experience in pediatric airway surgery and pediatric intensivists, bronchoscopists and pulmonologists with proven experience in the management of tracheostomized children |
| Primary variables | Tracheostomy indications, procedure technique, management of complications, specialized routine care, timing and technique for cannula exchange, timing of decannulation, patient and family orientations | Tracheostomy indications, cannula types, procedure technique, routine care components and techniques (aspiration, humidification), cannula changes and stoma care, management of complications, timing of decannulation, evaluation and treatment by other specialties (speech therapy, audiology), access to school |
| Outcomes | Not assessed | Not assessed |

and inner cannula cleaning were similar, the experimental group had higher peristomal skin care scores. In the final study, Sander et al. (2020) described a comprehensive program at a national referral hospital in Rwanda involving a low-fidelity simulation-based tracheostomy course for clinicians, an image-based tracheostomy care manual, and discharge tracheostomy care kits which cost 16 USD each [23]. Clinicians who underwent the simulation-based course demonstrated improved confidence levels in routine tracheostomy care, emergency situations, and in educating caregivers. The authors published the educational materials on OPENPediatrics, an open access online platform.

## Protocol development

Two articles from Brazil described the multidisciplinary process of developing standardized care protocols for adult and pediatric tracheostomy patients based on expert opinion (Table 4) [26, 27]. Guidelines were established based on consensus between multidisciplinary experts including otorhinolaryngologists, intensive care physicians, pulmonologists, oncologists, thoracic surgeons, and nurses. Primary components included indications for tracheostomy, tracheostomy technique, tracheostomy tube choice, management of routine and emergency tracheostomy care, criteria for decannulation, patient and family training, multidisciplinary care, and social integration. The outcomes of implementating these guidelines were not reported.

## Discussion

While QI interventions have led to evidence-based improvements in tracheostomy care in HICs, more efforts are needed to translate these lessons into comprehensive and contextualized models in LMICs. Existing evidence highlights an opportunity for QI given documented complications ranging from 7–22% among studies in this scoping review, comparable to that reported by hospitals in HICs [8]. This review highlights that tracheostomy care challenges in LMICs, where the majority of the world's population resides, are distinct compared to those experienced in HICs. Therefore, interventions in LMICs need to be resource-appropriate and

sustainable in multiple domains. Lessons from broader global health efforts in the following five domains help to frame lessons from this scoping review.

### Intervention integration within the healthcare system

Context-specific interventions in LMICs must integrate within the existing health system and be led by multidisciplinary teams working within resource-constrained contexts. For example, standardized operating protocols must be tailored to existing clinical practices and tracheostomy education with evaluation should be integrated into routine training of healthcare professionals.

### Sustainability of infrastructure and equipment

The sustainability of programs in LMIC settings needs to be considered. Specifically, for discharge home care supplies, Sandler et al. (2020) demonstrated that tracheostomy care kits can be cost-effective (16 USD per kit in this case) and locally sourced in Rwanda [23]. The team also piloted manually operated suction pumps as opposed to electrically operated devices circumventing the challenge of inconsistent electrical supply. Relying on existing infrastructure and promoting local demand can bolster existing supply chains for sustainability.

The search for alternative medical devices and equipment that are efficacious but available at lower prices is a common challenge across resource-constrained health settings. For example, in South Sudan, a device for endometrial biopsies to screen for endometrial cancer is estimated to cost 9.35 USD and demonstrates equal sensitivity and specificity compared to the use of the more expensive alternative of a hysteroscope and its associated instruments (6500 USD) [28, 29]. Additional efforts into locally-sourced equipment and infrastructure will provide long-term sustainability of quality improvement interventions.

### Innovation in care delivery

Given challenges exacerbated by constrained resources, innovative approaches to care delivery can improve care and follow-up. Interventions such as mobile follow-up as developed in India allow for caregivers to access continued education and limits the financial burden associated with travel for follow-up in regions where mobile devices are ubiquitous [25]. Mobile technology has been shown effective in healthcare delivery across multiple diseases, leading to improved behaviors such as increased exercise for participants in both HICs and LMICs [8, 30–32].

### Tailored educational materials

Materials must be translated into the local language, and picture-based training materials can help bridge gaps in the setting of low literacy. Low fidelity simulations can also be successfully implemented for the training of caregivers and patients on home care following hospitalization [33]. Simulations developed by teams in Rwanda and Malawi, for example, demonstrate an ability to bolster existing knowledge with a new method using local programmatic capacity [21, 23]. Furthermore, models of training the trainer establish a structure to train future cohorts of health professionals [34].

### Social and financial support

Social and financial support is particularly critical in LMICs. Home tracheostomy care was noted to pose a significant burden to family caregivers in multiple studies [10–13]. Family unemployment has also been reported to increase for patients with chronic or complex medical conditions given caregiver burden, exacerbating financial stress in low-income households [35, 36].

Financial constraints for families who care for multiple chronic conditions have been a common barrier. Studies have detailed catastrophic out of pocket health care expenditures for households of people with mental health disorders in Ethiopia, [37] and hypertension or diabetes in Pakistan [38]. These studies across disease areas highlight that an intervention to improve tracheostomy care quality must consider the potential financial toxicity and burden for families which can be barriers to care and treatment adherence [39].

With the unique challenges faced by patients and caregivers in LMICs, this scoping review highlights the pressing need for a systematic approach to tracheostomy care QI, specifically combining baseline needs assessment, an intervention addressing tracheostomy care QI at multiple phases and locations of care, and outcomes evaluation. To share lessons, outcomes for tracheostomy care QI efforts need to be measured and disseminated. In this review, only one interventional study measured end clinical outcomes of time to decannulation and 30-day mortality rates [24]. Other studies measured the perception of training educational interventions and change in knowledge among clinicians and caregivers. Given the comprehensive span of tracheostomy care, outcomes can be divided into the following key categories:

- Implementation outcomes: success of adoption of a model

- Service outcomes: knowledge and skills gained by providers

- Patient outcomes: complication and readmission rates, patient satisfaction

Outcomes in LMICs also ought to assess socioeconomic burden as home tracheostomy care was noted to pose a burden to family caregivers in multiple studies [10–13].

Given the complex challenges associated with tracheostomy care in LMICs, an implementation framework is one approach to address the comprehensive nature of building a successful tracheostomy care program that can incorporate the GTC key drivers of QI in the implementation strategies (Fig 2) [40]. Broadly, an implementation science framework is a structured approach to translate established interventions, such as the Global Tracheostomy Collaborative principles [7], into new settings [41]. Importantly, it also considers multiple phases of care along the care continuum. In the setting of tracheostomy care, this includes both hospital (ICU and ward education of care providers, multidisciplinary team, caregiver involvement early on) and home-based care (training of caregivers for post-discharge care and longitudinal follow-up). Future studies may consider this framework in the development and evaluation of tracheostomy QI programs.

This scoping review has multiple limitations. First, there may be a publishing bias in that not all programs developed may be published in English literature. Secondly, other examples of relevant tracheostomy related experiences in LMICs may not have been published due to lack of institutional research support and/or cost prohibitive publishing fees. Finally, the heterogeneity of studies inhibits comparing study approaches and outcomes.

Given the ubiquity of tracheostomy care, improving the quality of tracheostomy care and airway safety is critical worldwide. More work is needed to systematically assess and standardize care particularly in LMICs which face unique challenges. More baseline studies are needed to understand the complications and challenges unique to health settings in LMICs. Additionally, evidence-based interventions such as the approach by the Global Tracheostomy Collaborative need to be contextualized to LMIC health systems. Ultimately, using these lessons, a framework for continued QI can be expanded to other aspects of clinical care beyond tracheostomy care.

## Conclusions

While tracheostomy care in resource-constrained settings is challenging and a dearth of literature exists on this topic, pioneering hospitals and medical teams across the world have built

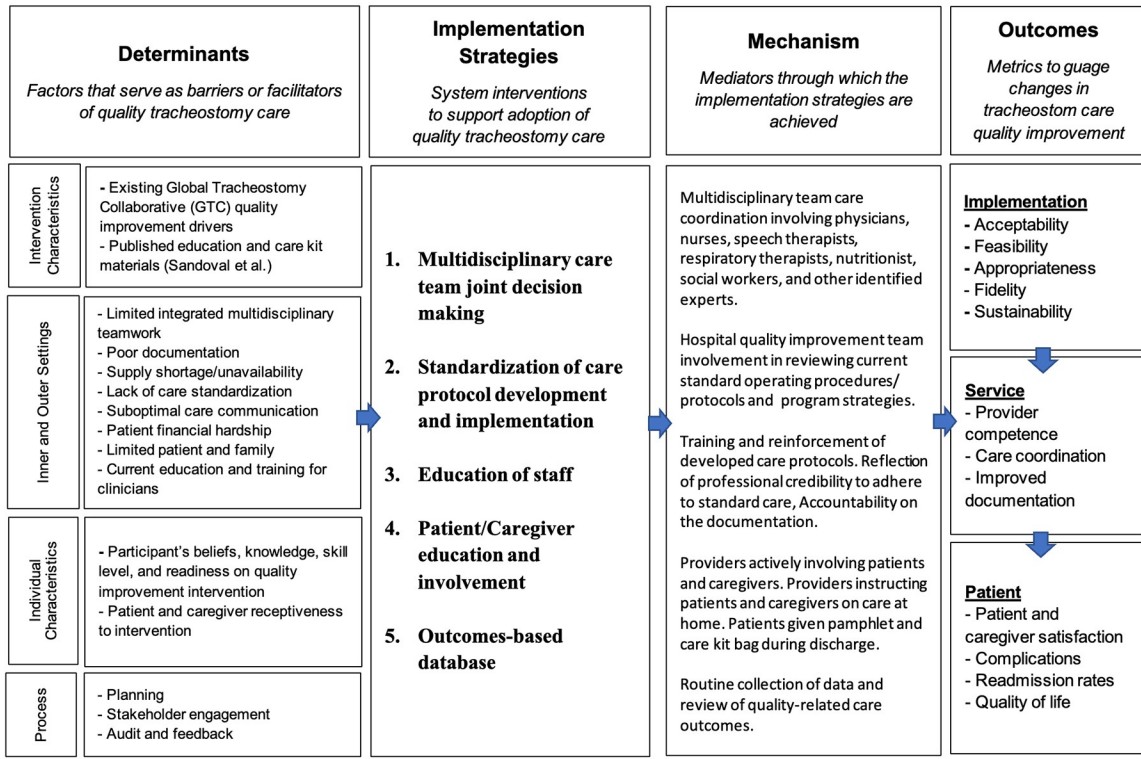

**Fig 2. A proposed implementation research logic model for framing tracheostomy care quality improvement.**

models for education and support of patients and their families. Overall, there is a need for more data collection to understand baseline complications. An implementation framework may furthermore serve as a systematic approach to implementing and adapting evidence-based approaches to improve tracheostomy care. This is critical as global efforts to expand essential surgeries are underway and tracheostomy care is a critical surgical procedure. Additionally, through efforts such as implementing tracheostomy care QI and educational interventions in tracheostomy care, this framework has the potential to be applied to other essential procedures on a global scale.

## Supporting information

**S1 Checklist. Preferred Reporting Items for Systematic reviews and Meta-Analyses extension for Scoping Reviews (PRISMA-ScR) checklist.**
(DOCX)

**S1 Appendix. Search strategy.**
(DOCX)

## Author Contributions

**Conceptualization:** Msiba Selekwa, Ivy Maina, Tiffany Yeh, Aslam Nkya, Isaie Ncogoza, Roger C. Nuss, Mary Jue Xu.

**Data curation:** Msiba Selekwa, Ivy Maina, Tiffany Yeh, Mary Jue Xu.

**Formal analysis:** Msiba Selekwa, Ivy Maina, Tiffany Yeh, Aslam Nkya, Isaie Ncogoza, Roger C. Nuss, Beatrice P. Mushi, Sumaiya Haddadi, Katherine Van Loon, Elia Mbaga, Willybroad Massawe, David W. Roberson, Nazima Dharsee, Baraka Musimu, Mary Jue Xu.

**Investigation:** Mary Jue Xu.

**Methodology:** Msiba Selekwa, Ivy Maina, Tiffany Yeh, Mary Jue Xu.

**Project administration:** Beatrice P. Mushi, Katherine Van Loon.

**Supervision:** Msiba Selekwa, Katherine Van Loon, Elia Mbaga, David W. Roberson.

**Visualization:** Tiffany Yeh.

**Writing – original draft:** Msiba Selekwa, Ivy Maina, Tiffany Yeh, Mary Jue Xu.

**Writing – review & editing:** Msiba Selekwa, Ivy Maina, Tiffany Yeh, Aslam Nkya, Isaie Ncogoza, Roger C. Nuss, Beatrice P. Mushi, Sumaiya Haddadi, Katherine Van Loon, Elia Mbaga, Willybroad Massawe, David W. Roberson, Nazima Dharsee, Baraka Musimu, Mary Jue Xu.

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
