## [Decision Letter · Decision Letter 0]

31 Aug 2023

PGPH-D-23-01326

Tracheostomy Care Quality Improvement in Low- and Middle-Income Countries: A Scoping Review

Dear Dr. Xu,

Thank you for submitting your manuscript to PLOS Global Public Health. After careful consideration, we feel that it has merit but does not fully meet PLOS Global Public Health’s publication criteria as it currently stands. Therefore, we invite you to submit a revised version of the manuscript that addresses the points raised during the review process.

We look forward to receiving your revised manuscript.

Kind regards,

Shahrzad Joharifard

Academic Editor

Journal Requirements:

2. Please provide separate figure files in .tif or .eps format only and remove any figures embedded in your manuscript file. Please also ensure all files are under our size limit of 10MB.

3. Tables should not be uploaded as individual files. Please remove these files and include the Tables in your manuscript file as editable, cell-based objects. For more information about how to format tables, see our guidelines:

https://journals.plos.org/globalpublichealth/s/tables

Additional Editor Comments (if provided):

Reviewers' comments:

Reviewer's Responses to Questions

**Comments to the Author**

1. Does this manuscript meet PLOS Global Public Health’s publication criteria? Is the manuscript technically sound, and do the data support the conclusions? The manuscript must describe methodologically and ethically rigorous research with conclusions that are appropriately drawn based on the data presented.

Reviewer #1: Yes

Reviewer #2: Yes

2. Has the statistical analysis been performed appropriately and rigorously?

Reviewer #1: N/A

Reviewer #2: N/A

3. Have the authors made all data underlying the findings in their manuscript fully available (please refer to the Data Availability Statement at the start of the manuscript PDF file)?

Reviewer #1: Yes

Reviewer #2: Yes

4. Is the manuscript presented in an intelligible fashion and written in standard English?

Reviewer #1: Yes

Reviewer #2: Yes

5. Review Comments to the Author

Reviewer #1: The article is very relevant and needed. The authors have reviewed the existing literature very well.

I would like to advise one thing, the article is very lengthy and at some points this makes it less interesting for the readers, making the article little short and to the point will help improve quality.

Reviewer #2: General

This paper addresses an important gap in available literature regrading tracheostomy care in LMICs. A scoping review seems appropriate. Exclusion of non-English publications is a significant limitation given the areas of the world the authors are trying to address.

Introduction

The authors appear to extrapolate rates of procedure and complications from HICs to LMICs, as there are no citations to support their claims of complication rates in LMICs. Having some support for the incidence of tracheostomy and associated complications would help make this stronger. Are rates of tracheostomy in LMICs as high as in HICs? This is assumed based on the overall higher global population. If the authors are making this assumption, it should be explicit.

Results

The logic model is a a helpful figure and contribution to further studies. While the goal of the logic model is different than the Global Tracheostomy Collaborative drivers of quality improvement (as outlined by the authors), it would bring the paper together to frame the logic model within the goals of the GTC, or at least tie them together somehow. While the goals are admirable, it seems that more wheels are being invented which ultimately undermines efforts to achieve these things.

6. PLOS authors have the option to publish the peer review history of their article (what does this mean?). If published, this will include your full peer review and any attached files.

**Do you want your identity to be public for this peer review?** For information about this choice, including consent withdrawal, please see our Privacy Policy.

Reviewer #1: No

Reviewer #2: No

---

## [Decision Letter · Decision Letter 1]

10 Oct 2023

Tracheostomy Care Quality Improvement in Low- and Middle-Income Countries: A Scoping Review

PGPH-D-23-01326R1

Dear Dr. Xu,

We are pleased to inform you that your manuscript 'Tracheostomy Care Quality Improvement in Low- and Middle-Income Countries: A Scoping Review' has been provisionally accepted for publication in PLOS Global Public Health.

Best regards,

Shahrzad Joharifard

Academic Editor

Reviewer Comments (if any, and for reference):

Reviewer's Responses to Questions

**Comments to the Author**

1. If the authors have adequately addressed your comments raised in a previous round of review and you feel that this manuscript is now acceptable for publication, you may indicate that here to bypass the “Comments to the Author” section, enter your conflict of interest statement in the “Confidential to Editor” section, and submit your "Accept" recommendation.

Reviewer #1: All comments have been addressed

Reviewer #2: (No Response)

2. Does this manuscript meet PLOS Global Public Health’s publication criteria? Is the manuscript technically sound, and do the data support the conclusions? The manuscript must describe methodologically and ethically rigorous research with conclusions that are appropriately drawn based on the data presented.

Reviewer #1: Yes

Reviewer #2: Yes

3. Has the statistical analysis been performed appropriately and rigorously?

Reviewer #1: N/A

Reviewer #2: N/A

4. Have the authors made all data underlying the findings in their manuscript fully available (please refer to the Data Availability Statement at the start of the manuscript PDF file)?

Reviewer #1: Yes

Reviewer #2: Yes

5. Is the manuscript presented in an intelligible fashion and written in standard English?

Reviewer #1: Yes

Reviewer #2: Yes

6. Review Comments to the Author

Reviewer #1: It is a good article and need of the time. Hopefully this will stimulate further research on Tracheostomy Care QI IN LMIC

Reviewer #2: (No Response)

7. PLOS authors have the option to publish the peer review history of their article (what does this mean?). If published, this will include your full peer review and any attached files.

**Do you want your identity to be public for this peer review?** For information about this choice, including consent withdrawal, please see our Privacy Policy.

Reviewer #1: **Yes: **Muhammad Hassan Danish

Reviewer #2: No
